# Research on a Lightweight Arrhythmia Classification Model Based on Knowledge Distillation for Wearable Single-Lead ECG Monitoring Systems

**DOI:** 10.3390/s24247896

**Published:** 2024-12-10

**Authors:** Xiang An, Shiwen Shi, Qian Wang, Yansuo Yu, Qiang Liu

**Affiliations:** Academy of Artificial Intelligence, Beijing Institute of Petrochemical Technology, Beijing 102617, China; anxiang@bipt.edu.cn (X.A.); 2022540059@bipt.edu.cn (S.S.); 2023312203@bipt.edu.cn (Q.W.); liuq@bipt.edu.cn (Q.L.)

**Keywords:** arrhythmia classification, knowledge distillation (KD), electrocardiogram (ECG), microcontroller, embedded system, edge intelligence

## Abstract

Arrhythmias are among the diseases with high mortality rates worldwide, causing millions of deaths each year. This underscores the importance of real-time electrocardiogram (ECG) monitoring for timely heart disease diagnosis and intervention. Deep learning models, trained on ECG signals across twelve or more leads, are the predominant approach for automated arrhythmia detection in the AI-assisted medical field. While these multi-lead ECG-based models perform well in automatic arrhythmia detection, their complexity often restricts their use on resource-constrained devices. In this paper, we propose an efficient, lightweight arrhythmia classification model using a knowledge distillation technique to train a student model from a teacher model, tailored for embedded intelligence in wearable devices. The results show that the student model achieves 96.32% accuracy, which is comparable to the teacher model, with a remarkable compression ratio that is 1242.58 times smaller, outperforming other lightweight models. Enabled by the proposed model, we developed a wearable ECG monitoring system based on the STM32F429 Discovery kit and ADS1292R chip, achieving real-time arrhythmia detection on small wearable devices.

## 1. Introduction

Cardiovascular disease (CVD) death rates are progressively rising as the world population ages, with an estimated rise to 20.5 million in 2025 and 35.6 million in 2050, representing a 73.4% increase over time [1]. As a common and dangerous group of CVDs, arrhythmias mainly refer to abnormal heart rhythms, including conditions such as fast, slow, irregular, or uncoordinated heartbeats. Arrhythmias may lead to long-term damage to the heart or sudden death; therefore, accurate and timely diagnosis of arrhythmias is crucial for effective cardiovascular disease treatment. Electrocardiogram (ECG) is an important non-invasive diagnostic tool that detects cardiovascular disease or other abnormalities by recording fluctuations in the electrical activity of the heart. ECG mainly consists of P waves, QRS complexes, T waves, etc., which represent the process of atrial depolarization, ventricular depolarization, and repolarization, respectively. The health of the heart can be assessed through the counting and analysis of these waveforms [2].

Traditional arrhythmia diagnosis relies on a cardiologist’s analysis of the patient’s ECG waveforms. The diagnosis is highly dependent on the physician’s experience and expertise. Manual diagnosis is not only time-consuming but also prone to errors. Therefore, the use of automated analysis technology for real-time ECG monitoring can significantly improve the diagnostic efficiency of arrhythmia [3]. Currently, artificial intelligence-enhanced ECG classification algorithms are mainly divided into feature extraction-based classification algorithms and deep learning-based classification algorithms. Feature extraction-based algorithms extract features from the original ECG signals in the time domain (e.g., RR interval, QRS wave width), frequency domain (e.g., power spectral density), and nonlinear features (e.g., entropy, fractal dimension). They then use machine learning algorithms such as support vector machines [4], Random Forests [5], and K-nearest neighbors [6] for classification. However, manual feature extraction and selection are cumbersome, prone to errors in the extraction process, and easily subject to interference from environmental noise.

With the advancement of deep learning, an increasing number of researchers are utilizing deep learning models for arrhythmia classification. Deep learning is able to automatically discover useful features directly from raw data through a computational model consisting of multilayer nonlinear processing modules [7]. For instance, previous studies have employed convolutional neural networks (CNNs) [8], deep neural networks (DNNs) [9], Residual Neural Networks (ResNet) [10], AlexNet [11], VGGNet [12] for arrhythmia classification. Both single-lead and multi-lead ECG signals can be used as input data for training deep learning models. However, since multi-lead ECGs provide different views of the heart’s electrical activity, offering more comprehensive cardiac information, many studies prefer to use multi-lead ECG datasets to train their models [13,14,15]. This preference enhances the models’ accuracy and robustness, but it also means these models are typically resource-intensive, challenging their deployment on wearable devices with limited computational capabilities.

In practical ECG monitoring scenarios, particularly for patients with chronic heart disease requiring long-term, continuous monitoring, connecting monitoring devices to cloud servers and transmitting data in real time not only generates a large amount of energy consumption and communication overhead but also poses risks to data security and privacy. Therefore, there is a pressing need for a lightweight ECG classification model that can perform real-time ECG analysis on edge devices, ensuring efficient, secure, and reliable arrhythmia classification. In addition, wearable single-lead ECG devices are better suited for long-term ECG monitoring, as they minimize the number of electrodes required from ten to two, greatly simplifying the user experience and enhancing wearability. Consequently, there is a growing necessity for arrhythmia classification models that can accurately interpret single-lead ECG signals. However, the performance of arrhythmia classification models trained directly on single-lead ECG often fails to match the performance levels of multi-lead models. For example, Yildirim et al. [16] proposed a DNN model trained and tested on a publicly available 12-lead ECG dataset with over 10,000 patients. Their results showed that the classification accuracy achieved using single-lead data was 92.24%, while the 12-lead data improved the accuracy to 96.13%, an increase of 3.89%.

Therefore, this paper introduces a lightweight arrhythmia classification method based on knowledge distillation. Initially, a high-precision teacher model is trained on a large-scale multi-lead ECG dataset. Subsequently, knowledge distillation is employed to transfer the learned knowledge from the teacher model to a lightweight student model designed for single-lead ECG analysis, significantly reducing the model’s parameter count and computational complexity while preserving high performance. Ultimately, the student model is deployed on a resource-constrained microcontroller to achieve efficient, real-time arrhythmia classification. The main contributions of this study are as follows:

(1)Proposed a lightweight and high-performance arrhythmia classification model with significant compression: This introduced a knowledge distillation (KD)-based framework to transfer the “dark knowledge” from a multi-lead ECG teacher model to a single-lead ECG student model. The proposed approach significantly reduces model complexity while achieving a classification accuracy of 96.32%, comparable to the multi-lead teacher model, with an impressive compression ratio of 1242.58 times. Additionally, the model was rigorously validated on the Chapman–Shaoxing ECG dataset, and comparisons with existing lightweight models further demonstrated its superiority.(2)Enabled real-time deployment on resource-constrained devices: A wearable ECG monitoring system based on the STM32F429 microcontroller (manufactured by STMicroelectronics, Geneva, Switzerland) was developed, demonstrating the feasibility of real-time arrhythmia classification in resource-limited environments.

## 2. Related Works

### 2.1. Intelligent ECG Analysis

In the field of intelligent ECG analysis, the majority of research is conducted using publicly available datasets, such as the MIT-BIH dataset [17]. Acharya et al. [18] used a nine-layer CNN to classify 452,960 cardiac beat samples with 94% accuracy, utilizing the MIT-BIH Arrhythmia Database. Chen et al. [19] combined a CNN and a long short-term memory network (LSTM) to automatically classify six types of ECG signals: normal (N) sinus rhythm, atrial fibrillation (AFIB), ventricular bigeminy (B), pacing rhythm (P), atrial flutter (AFL), and sinus bradycardia (SBR), and the model achieved an accuracy of 99.32%. This study also employed the MIT-BIH Arrhythmia Database. Hua et al. [13] proposed a novel 1D CNN which achieved classification accuracy, sensitivity, and F1-score of 97.45%, 99.25%, and 97%, respectively, according to the five categories suggested by AAMI (normal beats, supraventricular premature beats, ventricular premature beats, fusion beats, and unclassifiable beats). Their research was conducted using the MIT-BIH Arrhythmia Database as well. Somaraju et al. [14] proposed an LSTM-based classification model that performs multiclassification on the MIT-BIH dataset with a classification accuracy of 99.1% for ventricular ectopic beat (VEB) detection and 98.4% for supraventricular ectopic beat (SVEB) detection. Falaschetti et al. [15] proposed a recurrent neural network (RNN)-based classification model, which achieved 86.98% accuracy for multiclassification on the MIT-BIH dataset. Most of the aforementioned studies utilized the MIT-BIH dataset, but this dataset, due to its age, has issues with data imbalance. Moreover, because the types of labels are limited and the data are segmented by cardiac beats, it is difficult to adequately validate the dataset in practical use [20]. In 2020, Chapman University and Shaoxing People’s Hospital established a 12-lead ECG database, referred to as the Chapman–Shaoxing 12-lead ECG Database [21]. Based on this dataset, Yildirim et al. developed a deep neural network model combining feature representation learning and sequence learning for classifying 12-lead ECG signals from Chapman ECGs, and the results showed that the model achieved 92.24% accuracy in reduced rhythm category (seven types) scenarios, and 96.13% accuracy in combined rhythm category (four main types) scenarios [16]. Meqdad et al. [22] introduced an innovative 12-lead ECG signal fusion strategy utilizing evolutionary CNN trees, integrating frequency analysis, signal processing, and evolutionary computation methodologies. This technique attained an average accuracy of 97.60% on the Chapman ECG dataset, offering an enhanced strategy for arrhythmia diagnosis. Yoon et al. [23] devised a bimodal CNN model for the categorization of cardiovascular illness, attaining an AUC of 0.994 and an accuracy of 95.74% on the Chapman ECG dataset using the co-training of ECG grayscale pictures and scalograms.

Despite major advancements in accuracy, arrhythmia classification models utilizing multi-lead ECG data generally necessitate considerable computing resources, hence restricting their implementation to cloud-based processing. The increasing needs for real-time performance and data security in healthcare monitoring have compelled academics to investigate the implementation of complicated arrhythmia monitoring models nearer to the data source.

### 2.2. Edge Computing in Health Monitoring

The AI-based ECG analysis models mentioned above typically require computers or cloud platforms for their operation. However, with the advancement of wearable devices and the emergence of the concept of edge intelligence [24], the development of artificial intelligence models for wearable devices has piqued the interest of some researchers. By using edge intelligence, data processing is brought to the edge device, thereby enhancing real-time analytical capabilities, reducing latency, saving communication cost, and providing an improved user service experience [25]. Vimal et al. [26] designed a fall detection system based on Internet of Things (IoT) and edge computing, which reached 98% accuracy by deploying a deep convolutional neural network. Abdellatif et al. [27] designed an automated patient monitoring system using edge computing and blockchain technology. Real-time electrocardiogram (ECG) monitoring and analysis technology based on edge intelligence have also attracted considerable research interest due to their potential to transform healthcare delivery. Joukhadar et al. [28] developed portable devices utilizing Raspberry Pi for the real-time monitoring of heart valve disorders by automated segmentation and artificial neural networks. Likewise, Alfarhan et al. [29] employed support vector machines (SVMs) in wireless cardiac anomaly monitoring systems based on Raspberry Pi to categorize various electrocardiogram (ECG) signals. Sai et al. [30] implemented their custom-trained deep learning models on the NVIDIA Jetson Nano development kit for the categorization of arrhythmias utilizing ECG signals. Moreover, Zhang et al. [31] integrated ARM and FPGA to facilitate deep learning model inference, with the ARM processor executing the complete algorithm and the FPGA markedly enhancing inference speed via parallel processing. While this research illustrates the promise of using Raspberry Pi, NVIDIA Jetson Nano, and the ARM + FPGA combo for arrhythmia monitoring, these systems encounter some restrictions in wearable device applications.

Although Raspberry Pi and NVIDIA Jetson Nano exhibit substantial processing prowess, their considerable size and elevated power consumption render them impractical for prolonged wearable applications. The combination of ARM with FPGA provides enhanced performance; nevertheless, its high cost limits widespread adoption. This study utilized the STM32F429 microcontroller (manufactured by STMicroelectronics, Geneva, Switzerland), based on the ARM Cortex-M4 architecture, which supports floating-point operations appropriate for implementing generic neural networks. Its minimal power consumption, compact dimensions, lightweight construction, and cost efficiency render it optimal for the development of wearable arrhythmia monitoring systems.

### 2.3. Lightweight Models for Arrhythmia Classification

Due to the restricted processing resources of wearable devices, deploying models on these devices requires model optimization for reduced size and complexity. In prior studies, lightweight models were either explicitly developed for training or attained by methods such as knowledge distillation, pruning, and quantization. MobileNets [32] is a prevalent architecture-based lightweight model that minimizes the amount of model parameters by employing depthwise separable convolution. Khan et al. [33] introduced a Single-Shot Detection (SSD) MobileNet v2 deep neural network architecture for the identification of cardiac anomalies in ECGs, with an accuracy of 98%; however, this approach primarily focuses on the detection of 2D visual representations of ECG signals. Knowledge distillation (KD) [34] is a strategy for model compression and knowledge transfer that conveys information from a complicated, large model (teacher) to a simpler, smaller model (student). Sepahvand et al. [35] introduced a technique employing knowledge distillation to bridge the disparity between multi-lead ECG signal arrhythmia classification models and single-lead ECG signal models. Their student model had just a 0.81% decline in accuracy relative to the high-performing teacher model, yet the model size was decreased by around 262.18 times. Model pruning is a method that lowers the model’s size and computational expense by eliminating superfluous components of a neural network while preserving accuracy. Liu et al. [36] employed pruning strategies in arrhythmia classification, achieving a 47.6% reduction in model parameters and a 49.1% decrease in FLOPS (floating-point operations per second), while enhancing classification accuracy to 94.8%. Model quantization [37] is a method that transforms the parameters of deep learning models from high precision to lower precision to diminish storage demands and computational complexity, while striving to maintain performance. Wu et al. [38] achieved a 2.58-fold acceleration and a 61.27% decrease in energy usage in their heartbeat classification model by the implementation of weight quantization and parameter accuracy reduction. Although lightweight methods such as pruning, quantization, and knowledge distillation are highly effective in reducing model size and computational complexity, each approach has its own limitations in arrhythmia classification tasks. For instance, pruning can significantly reduce model complexity but often results in the loss of critical information, which may lead to a decline in precision and an increase in false positive rates. Quantization reduces the precision of model parameters to lower computational demands, but this often causes a noticeable drop in accuracy. Knowledge distillation, on the other hand, provides a balanced approach, but the performance of the student model heavily depends on the teacher model’s quality. Among these methods, this study adopts the knowledge distillation approach, as it achieves a better balance between model size, computational efficiency, and classification performance. By transferring “dark knowledge” from a high-performing teacher model to a lightweight student model, the proposed method effectively addresses the limitations of pruning and quantization while maintaining high precision and accuracy.

## 3. Materials and Methods

### 3.1. Datasets

In this study, we utilized the Chapman–Shaoxing 12-lead ECG database for training our arrhythmia classification model. The dataset contains ECG recordings from 10,646 patients, sampled at 500 Hz, and has been annotated by professional experts for 11 common heart rhythms and 67 additional cardiovascular conditions. Each patient’s record contains a 10 s ECG, saved in CSV format, which includes both raw and denoised ECG data. In the experiment, we ultimately used ECG recordings from 10,588 patients, as the presence of zero values and missing lead data in some recordings compromised their usability. Furthermore, to ensure that the trained model can be used in wearable ECG monitoring devices that support a sample rate of 250 Hz, we resampled the dataset to meet the input requirements of the model. We then performed Z-score normalization to enhance model performance and training efficiency. The Z-score normalization formula is shown in Equation (1):(1)$$Z=\frac{X-\mu }{\sigma }$$
where X is the original data, μ is the mean, and σ is the standard deviation. The dataset includes 11 rhythm classifications, although some of these categories include minimal cases. In light of the training data’s balance, we implemented the 4-class classification method suggested by the dataset developers. Atrial fibrillation (AFIB) and atrial flutter (AF) were consolidated into a singular AFIB category, while several supraventricular tachycardia-related rhythms (SVT—supraventricular tachycardia, AT—atrial tachycardia, SAAWR—sinus atrium to atrial wandering rhythm, ST—sinus tachycardia, AVNRT—atrioventricular node re-entrant tachycardia, AVRT—atrioventricular re-entrant tachycardia) were amalgamated into a comprehensive supraventricular tachycardia (GSVT) category. Sinus bradycardia (SB) was retained as a distinct category, but sinus rhythm (SR) and sinus arrhythmia (SI) were consolidated into a singular SR classification. The SR category in this study represents normal, healthy ECG signals, encompassing individuals with no arrhythmias or other abnormal cardiac rhythms. This ensures that the four-class classification framework already includes a distinct category for healthy signals. This integration facilitates more concentrated study while maintaining essential differences across beats. Ultimately, 70% of each integrated class was allocated to the training set, 20% to the validation set, and 10% to the test set, as seen in Table 1.

### 3.2. Knowledge Distillation

Knowledge distillation is a method that facilitates the transfer of knowledge from a complex teacher model to a simpler student model, and it has gained significant traction in the domain of deep learning in recent years. The fundamental concept involves utilizing the “soft labels” produced by the teacher model to inform the training process of the student model. Soft labels represent the probability distributions of categories produced by the teacher model, offering more comprehensive information than traditional “hard labels,” including insights into the similarities among various categories. Hard labels are the ground truth labels provided in the dataset. In this study, the arrhythmia classification task involves four categories: atrial fibrillation (AFIB), general supraventricular tachycardia (GVST), sinus bradycardia (SB), and sinus rhythm (SR). AFIB is labeled as 0, GVST as 1, SB as 2, and SR as 3. These labels represent the true class of each sample and are deterministic, guiding the student model to correctly classify the training samples. Soft labels, on the other hand, are the probability distributions generated by the teacher model for each sample. For example, the teacher model may output a soft label such as [0.8, 0.1, 0.05, 0.05], indicating an 80% probability for AFIB, a 10% probability for GVST, and so on. Unlike hard labels, which indicate the true class, soft labels capture richer inter-class information. In knowledge distillation, a large and complex teacher model is initially trained using hard labels. Subsequently, the trained teacher model is employed to generate predictions on the training data, resulting in the acquisition of soft labels. According to Hinton et al. [34] the calculation formula for soft labels is a variant of the Softmax function, where a temperature parameter T is introduced to control the degree of softening, as shown in Equation (2):(2)$$q_{i}=\frac{\exp\left(\frac{z_{i}}{T}\right)}{\sum_{j}\exp\left(\frac{z_{j}}{T}\right)}$$
where $z_{i}$ is the logit for class i , and T is the temperature parameter that controls the smoothness of the probability distribution. A higher temperature T produces softer probabilities, allowing the student model to better learn from the teacher model’s output distribution.

Next, we proceed to define a compact and straightforward student model, which is trained concurrently with the soft labels derived from the teacher model and the hard labels from the dataset. The training objective of the student model is to minimize two categories of losses:
(1)Distillation Loss: This metric quantifies the disparity between the predictions of the student model and the soft labels provided by the teacher model, facilitating the student’s acquisition of knowledge from the teacher. The measurement of distillation loss typically employs Kullback–Leibler (KL) divergence, as indicated in Equation (3):
(3)$$L_{\mathrm{distill}}=\mathrm{KL}(q^{T}∥p^{T})$$
where $q^{T}$ is the soft label from the teacher model, $p^{T}$ is the predicted probability distribution from the student model, and T is the temperature parameter.

In this context, the soft labels correspond to the teacher model’s outputs, while the predicted output pertains to the student model. The parameter *T* serves to regulate the “softness” of the soft labels.

(2)Cross-Entropy Loss: This measures the difference between the student model’s predictions and the true hard labels, ensuring that the student model maintains classification accuracy. The cross-entropy loss is shown in Equation (4): 

(4)$$L_{CE}=-\sum_{i}y_{i}\log(p_{i})$$
where $y_{i}$ represents the true hard label, and $p_{i}$ represents the predicted output of the student model.

The total loss of the student model is the weighted sum of the distillation loss and the cross-entropy loss, as shown in Equation (5):(5)$$L_{\mathrm{total}}=\alpha L_{\mathrm{CE}}+(1-\alpha )L_{\mathrm{distill}}$$
where α is a weight parameter that balances the two losses. By jointly optimizing these two losses, the student model can learn from the teacher model while also paying attention to the true labels, ultimately achieving a balance between accuracy and efficiency.

The pseudocode in Algorithm 1 describes the knowledge distillation process for training a lightweight arrhythmia classification model. A complex teacher model is trained on a 12-lead ECG dataset to learn complex spatial and temporal features. The knowledge learned by the teacher model is then transferred to a simpler student model designed for single-lead ECG classification. During the distillation process, the teacher model generates soft labels, which are combined with hard labels to compute the total loss, balancing classification accuracy and knowledge transfer.
Algorithm 1: Knowledge Distillation for Lightweight Arrhythmia Classification$\begin{array}{l} 1\;\;\text{This algorithm trains a lightweight student model using knowledge distillation from a pre-trained}\\ \;\text{teacher model to classify single-lead ECG signals efficiently.}\\ \;Input:\;Pre-trainset\;D_{\text{multi}}\;(12\text{-lead}\;\text{ECG}\;\text{dataset}),\\ \;\;\;\text{Downstreamset}\;D_{\text{single}}\;(\text{single-lead}\;\text{ECG}\;\text{dataset}\;),\\ \;\;\;\text{Pre-trained}\;\text{teacher}\;\text{model}\;T,\;\text{lightweight}\;\text{student}\;\text{model}\;S;\\ \;Output:\;\text{Trained}\;\text{lightweight}\;\text{student}\;\text{model}\;S\\ 2\;\;\text{Pre-training Teacher Model:}\\ 3\;\;for\;each\;x\in D_{\text{multi}}\;do\\ 4\;p_{T}\leftarrow T(x)\;\;\;\;\;//Teacher\;model\;processes\;12-lead\;ECG\;data\\ 5\;\text{Train}\;T\;\text{using}\;\text{cross-entropy}\;\text{loss}\;L_{\text{CE}};\\ 6\;\text{Update}\;T\;\text{via}\;\text{backpropagation};\\ 7\;\text{Knowledge Distillation:}\\ 8\;for\;each\;x\in D_{\text{single}}\;do\\ 9\;\;\;p_{T}\leftarrow \text{Softmax}(T(x)/T)\;\;\;\;//Teacher\;model\;generates\;soft\;labels\\ 10\;\;p_{S}\leftarrow \text{Softmax}(S(x)/T)\;\;\;\;\;//Student\;model\;predicts\;output\\ 11\;\;L_{\text{distill}}=\mathrm{KL}(p_{T}||p_{S})\;\;\;\;\;\;//Compute\;Distillation\;Loss\;(Eq.3)\\ 12\;\;L_{\text{class}}=\text{CrossEntropy}(y,p_{S})\;\;\;//Compute\;Classification\;Loss\;(Eq.4)\\ 13\;\;L_{\mathrm{total}}=\alpha L_{\text{class}}+(1-\alpha )L_{\text{distill}}\;\;\;//Combine\;Losses\;(Eq.5)\\ 14\;\;\text{Update}\;S\;u\text{sin}g\;L_{\text{total}}\;\text{via}\;\text{backpropagation};\\ 15\;return\;\text{Trained}\;\text{lightweight}\;\text{student}\;\text{model}\;S; \end{array}$

#### 3.2.1. Teacher Model

The teacher model typically exhibits greater complexity and depth in its structure compared to the student model. The teacher model can select from various classic convolutional neural network architectures depending on the task. This study employs a tailored convolutional recurrent neural network architecture as the teacher model, as shown in Figure 1. The teacher model comprises two primary components: global feature extraction and sequence feature extraction. The global feature extraction component employs ResNet-inspired convolutional modules alongside Squeeze-and-Excitation (SE) blocks [39]. Convolutional layers extract local features from ECG signals, while residual connections mitigate optimization challenges in deep networks. Additionally, SE blocks adaptively adjust the weights of each feature channel. The sequence feature extraction component employs the LSTM (Long Short-Term Memory) module to identify long-term temporal dependencies in ECG signals. The input to the teacher model is initially processed by a convolutional layer for feature extraction, followed by four ResNet blocks for deeper feature extraction. Each ResNet block comprises two convolutional layers, batch normalization layers, SE blocks, and residual connections. The extracted feature vectors are subsequently input into two layers of LSTM for the purpose of sequence modeling. The output of the LSTM is processed through fully connected layers followed by a Softmax layer for classification prediction. The teacher model is trained on 12-lead ECG data, effectively leveraging the complementary information across various leads to achieve robust representation capabilities.

To evaluate the contribution of each component in the teacher model (i.e., ResNet blocks, LSTM layers, and SENet blocks), we conducted an ablation study. In this study, we compared the complete teacher model with three modified versions: (1) the teacher model without ResNet blocks, (2) the teacher model without LSTM layers, and (3) the teacher model without SENet blocks. As shown in Table 2, the complete teacher model achieved the highest overall accuracy (97.50%), with superior sensitivity, precision, specificity, and F1-score across all classes. In contrast, removing the ResNet blocks (Table 3) resulted in a drop in accuracy to 96.50%, with significant declines in precision and specificity, particularly for the GSVT and AFIB classes. Similarly, removing the LSTM layers (Table 4) impaired the model’s ability to capture temporal dependencies in ECG signals, reducing the accuracy to 97.02% and significantly lowering sensitivity. Finally, removing the SENet blocks (Table 5) caused the accuracy to drop to 96.69%, with noticeable misclassifications in the AFIB and SR classes. The confusion matrices corresponding to the different configurations further highlight these performance differences. The confusion matrix for the complete teacher model (Figure 2a) demonstrates the highest classification accuracy, with minimal misclassifications across all classes. Only a few errors were observed, such as occasional misclassifications between AFIB and GSVT or between SR and SB. In contrast, after removing the ResNet blocks (Figure 2b), the confusion matrix shows a significant increase in misclassifications between AFIB and GSVT, underscoring the importance of ResNet blocks for robust spatial feature extraction. Similarly, removing the LSTM layers (Figure 2c) resulted in a higher misclassification rate between SR and SB, highlighting the critical role of LSTM layers in capturing temporal dependencies. Finally, the confusion matrix for the teacher model without SENet blocks (Figure 2d) reveals a decline in performance for the AFIB and SR classes, with more frequent misclassifications, emphasizing the importance of SENet blocks in feature recalibration. These results collectively validate the necessity of each component in the teacher model for achieving excellent classification performance.

#### 3.2.2. Student Model

The student model’s structure is simpler and more lightweight than that of the teacher model, facilitating adaptation to resource-constrained deployment environments. This study presents the design of a small convolutional neural network serving as the student model, as shown in Figure 1. The student model utilizes single-lead (lead II) ECG data as input. In comparison to the teacher model, the student model comprises two convolutional layers, one fully connected layer, and a Softmax output layer. The initial convolutional layer conducts feature extraction, while the subsequent convolutional layer enhances the extraction of high-level features. ReLU activation functions are applied subsequent to the convolutional layers to incorporate nonlinearity. The convolutional feature maps are ultimately flattened and input into the fully connected layer for classification purposes. The student model produces a probability distribution across four rhythm categories.

### 3.3. Assessment Indicators

To evaluate the performance of arrhythmia classification models, we utilize several standard evaluation metrics in machine learning: accuracy, sensitivity (recall), precision, F1-score, and specificity. These metrics are derived from the confusion matrix, which juxtaposes predicted labels with true labels, offering a visual representation of the model’s classification performance [40]. The confusion matrix comprises four fundamental components: true positives (TPs), true negatives (TNs), false positives (FPs), and false negatives (FNs). TP denotes the count of correctly classified positive samples, TN indicates the count of correctly classified negative samples, FP refers to the count of negative samples misclassified as positive, and FN signifies the count of positive samples misclassified as negative. The evaluation metrics are computed based on these elements as follows:(6)$$\mathrm{Accuracy}=\frac{TP+TN}{\mathrm{TP}+\mathrm{TN}+\mathrm{FP}+\mathrm{FN}}$$
(7)$$\mathrm{Sensitivity}=\frac{TP}{TP+FN}$$
(8)$$\mathrm{Precision}=\frac{TP}{TP+FP}$$
(9)$$F1-\mathrm{Score}=\frac{2\times \mathrm{Precision}\times \mathrm{Sensitivity}}{\mathrm{Precision}+\mathrm{Sensitivity}}$$
(10)$$\mathrm{Specificity}=\frac{TN}{FP+TN}$$

Accuracy evaluates the comprehensive correctness of the model’s predictions. Sensitivity (recall) quantifies the model’s capacity to accurately identify positive samples, whereas precision assesses the accuracy of the model’s positive predictions. The F1-score represents the harmonic mean of precision and recall, offering a balanced assessment of model performance. Specificity quantifies the model’s capacity to accurately identify negative samples. Considering these metrics collectively allows for a comprehensive assessment of the performance of arrhythmia classification models.

### 3.4. Result

Model training and validation were conducted on a workstation equipped with an AMD Ryzen 7 6800H CPU, 32 GB of RAM (manufactured by AMD, Santa Clara, USA), and an NVIDIA RTX 3060 GPU with 6 GB of VRAM (manufactured by AMD, Santa Clara, USA). We employed the Adam optimizer for both the teacher and student models throughout the training procedure. The student model underwent training for 200 epochs with a batch size of 64, an initial learning rate of 0.01, and a minimum learning rate of 0.00001. The learning rate was decreased by a factor of 0.1 if the validation accuracy did not improve for 10 consecutive epochs. Training was halted either upon the completion of all epochs or when the learning rate dropped to the minimum threshold.

Figure 3a illustrates the accuracy curve of the student model on the training and validation datasets during the process, while Figure 3b shows the corresponding loss curve. The model’s accuracy on the training set rises as the loss diminishes, indicating successful learning from the training data. Furthermore, the accuracy curve on the validation set mirrors that of the training set, indicating the model’s strong generalization capability and resilience to unfamiliar data. The student model attains a notable accuracy of 96.32% on the validation set, underscoring the efficacy of the knowledge distillation method in enhancing the performance of lightweight models.

To examine the effect of knowledge distillation on model performance, we compare the confusion matrices and evaluation metrics of the student model both with and without knowledge distillation, as seen in Figure 4a,b and Table 6 and Table 7. The findings unequivocally illustrate the advantages of information distillation, as the model using this method displays superior accuracy, sensitivity, precision, F1-score, and specificity across all arrhythmia categories in comparison to the model trained without knowledge distillation. The total accuracy of the student model increases from 94.44% to 96.32% with the use of knowledge distillation, while preserving a small model size appropriate for deployment on resource-limited devices. According to the results, the efficacy of the proposed knowledge distillation method in creating lightweight and highly accurate arrhythmia classification models is evident. By leveraging insights from a large, complex teacher model and transferring them to a smaller student model, we can significantly enhance the performance of the student model while keeping computational requirements. This approach has substantial implications for the development of real-time, edge computing-based arrhythmia monitoring systems, as it allows for the deployment of robust classification models on resource-constrained wearable devices.

## 4. Deployment of Lightweight Model

### 4.1. Hardware Structure

The real-time ECG monitoring system developed in this study primarily comprises an STM32F429 microcontroller (manufactured by STMicroelectronics, Geneva, Switzerland) and an ADS1292R ECG signal acquisition module (manufactured by Texas Instruments, Dallas, USA). The STM32F429 is a microcontroller characterized by its ARM Cortex-M4 core, offering 2 MB of flash memory, 256 KB of RAM, and a maximum clock frequency of 180 MHz. It integrates DSP instructions and an FPU unit, rendering it suitable for executing complex signal processing algorithms. The ADS1292R is a low-power, high-resolution chip designed for ECG signal acquisition, featuring an integrated programmable gain amplifier (PGA), a 12-bit analog-to-digital converter (ADC), and a digital filter. This chip effectively mitigates power supply interference and baseline drift, enhancing signal quality.

Figure 5 illustrates the structural block diagram and monitoring procedure of the wearable ECG monitoring system developed in this work.

The ADS1292R interfaces with the STM32F429 via SPI, relaying the obtained raw ECG signals to the microcontroller. The STM32F429 preprocesses the signals through denoising, filtering, and normalization, subsequently employing the deployed student model for real-time classification of ECG signals. Classification results may be communicated to a host computer through a UART (Universal Asynchronous Receiver-Transmitter) interface or presented directly on an LCD screen. The system incorporates a Bluetooth module, facilitating the wireless transmission of ECG data and analysis results to a smartphone or cloud server for remote monitoring and storage.

### 4.2. Model Deployment Based on STM32CubeAI

The trained student model was implemented on a resource-limited microcontroller using the STM32CubeAI toolkit (developed by STMicroelectronics, Geneva, Switzerland) from STMicroelectronics. This toolkit facilitates the deployment of deep learning models from many frameworks, such as TensorFlow and PyTorch, onto STM32-series microcontrollers. STM32CubeAI streamlines the deployment of models on edge devices by providing functionalities such as model conversion, optimization, and code creation. Figure 6 demonstrates the integration of the student model into the STM32F429 development board using STM32CubeAI. The real-time identification of arrhythmias on wearable devices using this model necessitates procedures including signal gathering, model analysis, and code creation. The teacher and student models were developed in TensorFlow and then compressed using knowledge distillation. Following model analysis, the STM32CubeAI toolkit produced optimized C code for the STM32F429 development board. Electrocardiogram (ECG) data from the ADS1292R module were used for the real-time categorization of four arrhythmia types. The STM32CubeAI Keras-to-C converter was used to convert the trained student model from its native Keras format into C code throughout the conversion process. The produced model code was included into the project, built, and uploaded to the STM32F429 development board with the STM32Cube IDE environment. During model inference, we used the optimized libraries offered by STM32CubeAI, including CMSIS-NN, to improve the performance of essential operations such as convolution and pooling.

### 4.3. Performance Evaluation

Table 8 presents a comparison of the performance of various classification models, including the proposed student model, when implemented on the STM32F429 microcontroller (manufactured by STMicroelectronics, Geneva, Switzerland) for arrhythmia detection. The metrics evaluated include test accuracy, the count of multiply–accumulate operations (MACs), read-only memory (ROM) utilization, random-access memory (RAM) utilization, and inference duration. The models compared in Table 8, including RNN, LSTM, and CNN_LSTM, are widely recognized and frequently used architectures for ECG-based arrhythmia classification, particularly in wearable device applications. RNN and LSTM are known for their ability to handle sequential data, capturing temporal dependencies in ECG signals. CNN_LSTM combines convolutional layers for spatial feature extraction with LSTM layers for temporal modeling, making it a popular hybrid architecture. These models were selected as benchmarks to evaluate the proposed student model’s performance in both accuracy and computational efficiency.

The results indicated that the student model exhibits superior overall performance on the resource-constrained embedded platform, surpassing other models in accuracy, model size, and inference speed. Specifically, the proposed student model achieves the highest test accuracy (96.32%), surpassing CNN_LSTM (94.71%) and significantly outperforming RNN (92.54%) and LSTM (83.02%). This demonstrates the effectiveness of the proposed model in maintaining high classification performance.

In terms of computational efficiency, the proposed student model has the lowest MACs (84,184), ROM utilization (25.34 KiB), and RAM utilization (11.65 KiB), making it highly suitable for deployment on resource-constrained embedded platforms. By comparison, CNN_LSTM, while achieving competitive accuracy (94.71%), requires significantly higher computational resources, with MACs of 905,072 and ROM utilization of 48.70 KiB. Furthermore, the proposed model achieves the shortest inference time (19 ms), which is critical for real-time applications in wearable systems. This is a substantial improvement compared to the RNN (253 ms) and the LSTM (388 ms), highlighting the model’s practical advantages in edge computing scenarios. The enhanced performance is due to the implementation of knowledge distillation, allowing the student model to acquire an efficient feature representation from the extensive and intricate teacher model. The proposed student model can achieve high accuracy while substantially decreasing its size and computational complexity.

### 4.4. System Implementation

To evaluate the performance of the deployed lightweight arrhythmia classification model, we used an SKX-2000 ECG simulator (manufactured by Xuzhou Minsheng Electronics Co., Ltd., Xuzhou, China) to generate a diverse set of arrhythmia signals. These signals were input into the experimental hardware and displayed in real time on an LCD screen. Concurrently, the lightweight student model embedded in the MCU analyzed the incoming signals and presented classification results on the LCD screen. As shown in Figure 7, the implemented real-time ECG monitoring and arrhythmia detection system successfully achieved real-time ECG monitoring and arrhythmia classification. This system facilitates precise and efficient arrhythmia detection on edge devices, thereby potentially improving patient outcomes, lowering healthcare costs, and enhancing the overall quality of care. Its applicability could extend to remote healthcare monitoring, personalized medicine, and wearable technology.

## 5. Conclusions

This study presents a lightweight arrhythmia classification model via knowledge distillation, suitable for deployment on resource-constrained wearable devices for real-time ECG monitoring. The proposed approach first trains a high-accuracy teacher model using an extensive 12-lead ECG dataset, thereafter, transferring the acquired information to a compact student model tailored for single-lead ECG interpretation via knowledge distillation. Experimental findings indicate that the student model attains an accuracy of 96.32% on the test set, similar to the teacher model, with an impressive model compression ratio of 1242.58 times. Additionally, we deployed the proposed lightweight model on an STM32F429 microcontroller (manufactured by STMicroelectronics, Geneva, Switzerland) via the STM32CubeAI toolbox (developed by STMicroelectronics, Geneva, Switzerland). The implemented model exhibits enhanced performance relative to other prevalent arrhythmia classification models for accuracy, model size, and inference speed. Consequently, we devised a comprehensive wearable ECG monitoring system that incorporates ECG signal capture, real-time arrhythmia identification, and wireless data transfer functionalities. The system has been effectively evaluated using an ECG simulator and shown precise classification of diverse arrhythmia types in real time.

In general, the proposed method addresses many critical obstacles in the deployment of AI-driven arrhythmia detection on wearable devices. Initially, by using knowledge distillation, it enables the creation of highly accurate yet compact models that can operate effectively on microcontrollers with limited memory and processing capabilities. This eliminates the need for cloud-based processing, thereby reducing latency, battery consumption, and data security vulnerabilities associated with wireless data transfer. Secondly, the system supports single-lead ECG monitoring, which significantly enhances wearability and user experience compared to traditional multi-lead ECG systems. Despite using only a single lead, the model maintains superior classification efficacy by acquiring knowledge from a teacher model trained on multi-lead data.

Nonetheless, there are several limits and potential areas for enhancements in the future. One issue is that the proposed strategy requires a substantial amount of annotated multi-lead ECG data to train the teacher model, which may not always be readily available. To address this, we plan to explore unsupervised or semi-supervised learning approaches in future work. Furthermore, we aim to refine the model architecture and training techniques to further minimize the model size and enhance performance, thereby enabling deployment on even more resource-constrained devices.

## Figures and Tables

**Figure 1 sensors-24-07896-f001:**
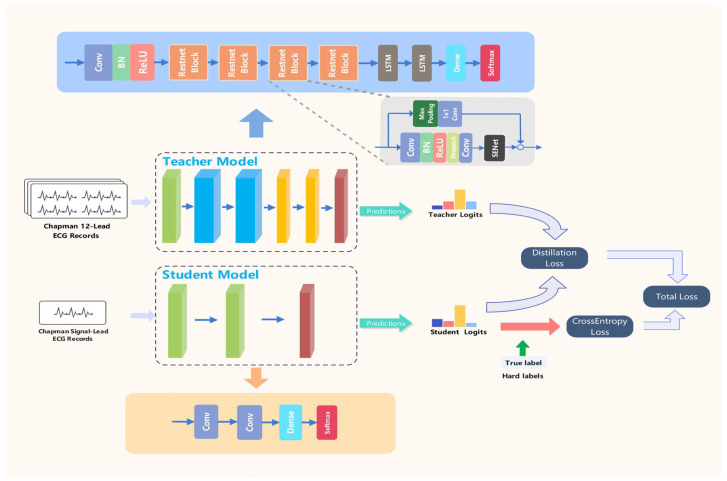
Model architecture and knowledge distillation process. It illustrates the knowledge distillation process, wherein the information acquired by the teacher model is efficiently conveyed to the student model, allowing it to attain superior performance with a more streamlined design.

**Figure 2 sensors-24-07896-f002:**
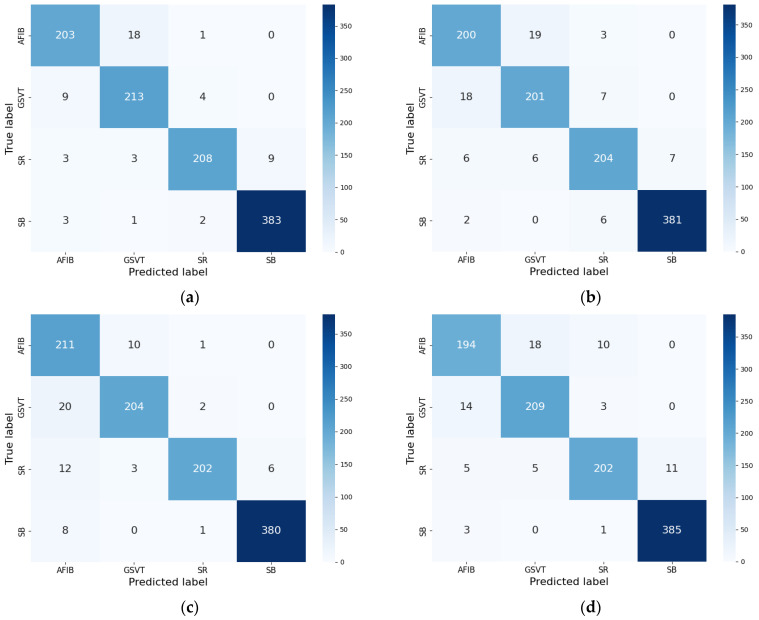
Confusion matrices for the teacher model under different configurations. (**a**) Confusion matrix for the complete teacher model; (**b**) confusion matrix for the teacher model without ResNet blocks; (**c**) confusion matrix for the teacher model without LSTM layers; (**d**) confusion matrix for the teacher model without SENet blocks.

**Figure 3 sensors-24-07896-f003:**
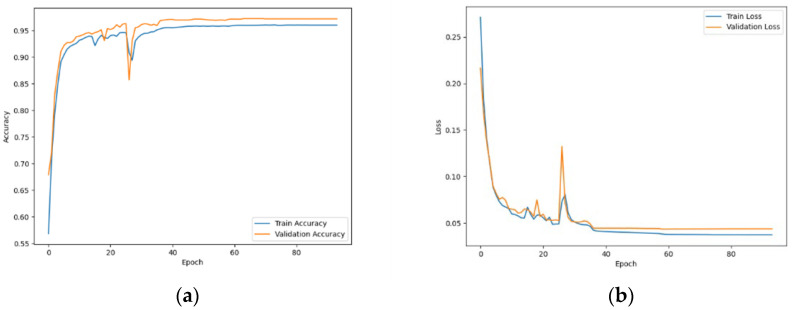
Training and validation curves for the student model. (**a**) Accuracy curve of the student model during training and validation; (**b**) loss curve of the student model during training and validation.

**Figure 4 sensors-24-07896-f004:**
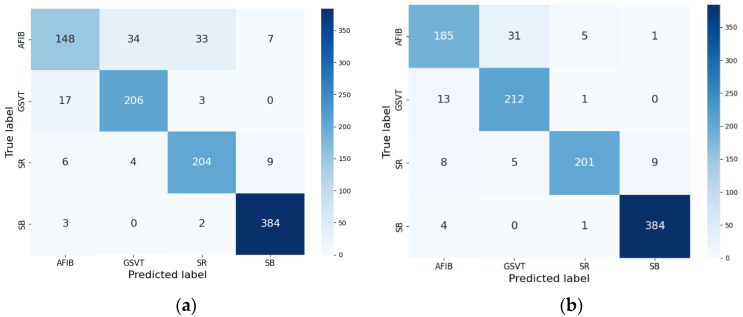
Confusion matrices for the student model without and with knowledge distillation. (**a**) Confusion matrix for the student model without knowledge distillation; (**b**) confusion matrix for the student model with knowledge distillation.

**Figure 5 sensors-24-07896-f005:**
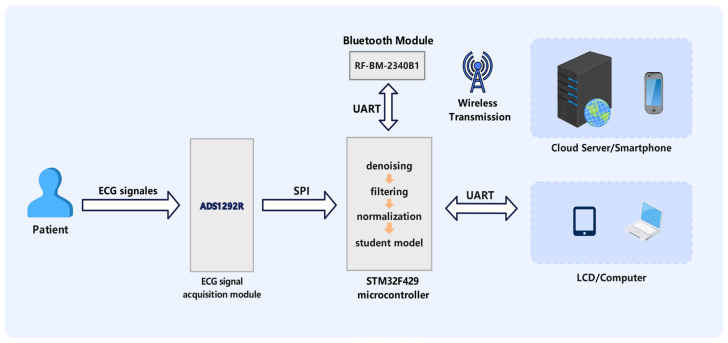
Wearable ECG monitoring system block diagram.

**Figure 6 sensors-24-07896-f006:**
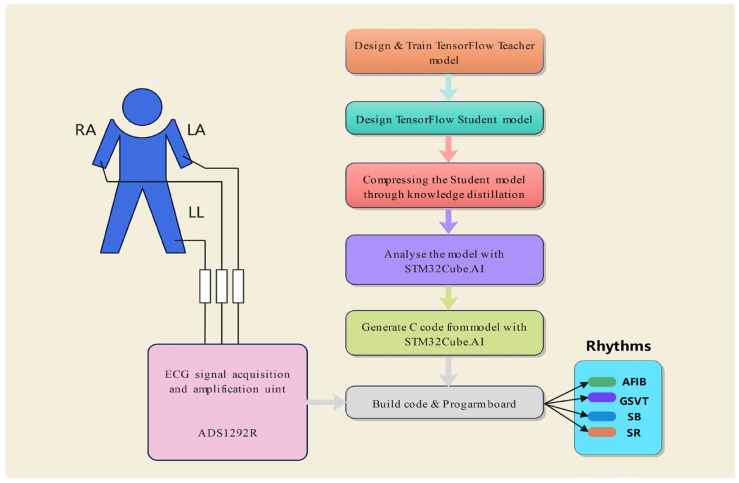
Process of integrating the student model into the STM32F4 board with STM32Cube.AI.

**Figure 7 sensors-24-07896-f007:**
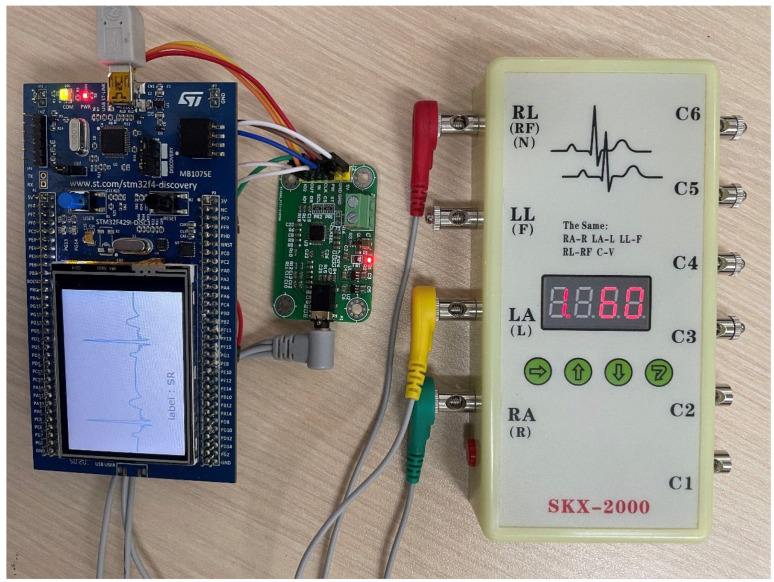
The real-time ECG monitoring and arrhythmia detection system.

**Table 1 sensors-24-07896-t001:** Rhythms categories and their corresponding sample counts.

**Merged Rhythms**	**New Class Name**	**Number of Total Samples**	**Number of Training Samples**	**Number of Testing Samples**	**Age, Mean ± STD**
AF+ AFIB	AFIB	2218	1983	235	72.92 ± 11.66
SVT + AT + SAAWR + ST+ AVNRT + AVRT	GSVT	2260	2061	199	55.51 ± 20.41
SB	SB	3888	3488	400	58.33 ±13.95
SR, SI	SR	2222	1997	225	50.89 ± 19.18
All		10,588	9529	1059	59.23 ± 17.97

**Table 2 sensors-24-07896-t002:** Class-based performance values for the complete teacher model.

**Classes**	**Sensitivity (%)**	**Precision (%)**	**Specificity (%)**	**F1-Score (%)**	**Accuracy (%)**
AFIB	91.44	93.12	98.21	92.27	96.79
GSVT	94.25	90.64	97.36	92.41	96.70
SB	98.46	97.70	98.66	98.08	98.58
SR	93.27	96.74	99.16	94.98	97.92
Overall	94.35	94.55	98.34	94.43	97.50

**Table 3 sensors-24-07896-t003:** Class-based performance values for the teacher model without ResNet blocks.

**Classes**	**Sensitivity (%)**	**Precision (%)**	**Specificity (%)**	**F1-Score (%)**	**Accuracy (%)**
AFIB	91.75	94.50	98.53	89.29	95.47
GSVT	88.94	88.94	97.00	88.94	95.28
SB	97.94	98.20	98.96	98.07	98.58
SR	91.48	92.73	98.09	92.10	96.70
Overall	92.52	93.59	98.14	92.10	96.50

**Table 4 sensors-24-07896-t004:** Class-based performance values for the teacher model without LSTM layers.

**Classes**	**Sensitivity (%)**	**Precision (%)**	**Specificity (%)**	**F1-Score (%)**	**Accuracy (%)**
AFIB	95.05	84.06	95.23	89.22	95.19
GSVT	90.27	94.01	98.44	92.10	96.70
SB	97.69	98.45	99.11	98.06	98.58
SR	90.58	98.06	99.52	94.17	97.64
Overall	93.39	93.64	98.07	93.38	97.02

**Table 5 sensors-24-07896-t005:** Class-based performance values for the teacher model without SENet blocks.

**Classes**	**Sensitivity (%)**	**Precision (%)**	**Specificity (%)**	**F1-Score (%)**	**Accuracy (%)**
AFIB	87.39	89.81	97.37	88.58	95.28
GSVT	92.48	90.09	97.24	91.27	96.23
SB	98.97	97.22	98.36	98.09	98.58
SR	96.70	93.52	98.33	92.03	96.70
Overall	93.88	92.66	97.82	92.49	96.69

**Table 6 sensors-24-07896-t006:** Class-based performance values for the model without knowledge distillation.

**Classes**	**Sensitivity (%)**	**Precision (%)**	**Specificity (%)**	**F1-Score (%)**	**Accuracy (%)**
AFIB	66.67	85.06	96.90	74.75	90.57
GSVT	91.15	84.43	95.44	87.66	94.53
SB	98.71	96.00	97.62	97.34	98.02
SR	91.48	84.30	95.46	87.74	94.62
Overall	87.00	87.45	96.36	86.87	94.44

**Table 7 sensors-24-07896-t007:** Class-based performance values for the model with knowledge distillation.

**Classes**	**Sensitivity (%)**	**Precision (%)**	**Specificity (%)**	**F1-Score (%)**	**Accuracy (%)**
AFIB	83.33	88.10	97.02	85.65	94.15
GSVT	93.81	85.48	95.68	89.45	95.28
SB	98.71	97.46	98.51	98.08	97.26
SR	90.13	96.63	99.16	93.27	98.58
Overall	91.50	91.92	97.59	91.61	96.32

**Table 8 sensors-24-07896-t008:** Performance comparison of classification models on STM32F429 embedded platform.

**Model**	**Test Accuracy**	**MACs**	**ROM (KiB)**	**RAM (KiB)**	**Inference Time (ms)**
RNN [41]	92.54%	2,784,240	43.84	67.69	253
LSTM [42]	83.02%	3,624,304	48.29	52.09	388
CNN_LSTM [43]	94.71%	905,072	48.70	35.99	93
Proposed Student Model	96.32%	84,184	25.34	11.65	19

## Data Availability

The data presented in this study is openly available in https://figshare.com/collections/ChapmanECG/4560497/2, Posted on 2019-11-30 - 05:03 authored by Jianwei Zheng.

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
