# Peer review of "Research on a Lightweight Arrhythmia Classification Model Based on Knowledge Distillation for Wearable Single-Lead ECG Monitoring Systems"

_sensors, 2024, doi:10.3390/s24247896_

Round 1

Reviewer 1 Report

Comments and Suggestions for Authors

This paper proposes a new method for lightweight arrhythmia classification based on knowledge distillation, designed for wearable single-lead ECG monitoring systems. However, several aspects require revision.

Firstly, regarding the deep learning model, it is overly simplistic, consisting of only four ResNet blocks and two LSTM blocks, lacking significant innovation. Moreover, the four-class classification is not entirely appropriate; a five-class classification, including a distinct category for healthy ECG signals, would be more reasonable. Furthermore, in the discussion of the mode’s classification results, there is a lack of comparative analysis. For example, a comparison with existing non-lightweight models should be provided, highlighting how the model improves computational speed with minimal loss in accuracy. Alternatively, a comparison with existing lightweight models should be made to show the improvement in accuracy achieved by the proposed model. Also, the references for the models being compared should be included in Table 4. In terms of results, K-fold cross-validation is recommended to avoid chance from splitting data only once.

Secondly, with respect to the hardware, Figure 6 is insufficiently comprehensive, as it should also present the classification results for both the four-class ECG signals and the healthy ECG signals.

Lastly, in terms of formatting, the resolution of the images in the manuscript is too low, likely due to screenshots; it is recommended to use vector graphics for improved clarity.

In addition, there are some minor concerns:

“Hua et al. [13] proposed a novel 1D CNN network which achieved classification accuracy, sensitivity, and F1 score of 97.45%, 99.25%, and 0.97%, respectively”. “0.97%” should be changed to “0.97” or “97%”.

Comments on the Quality of English Language

No

Author Response

Dear Reviewer,

Thank you very much for your valuable feedback on our manuscript. We have carefully addressed all your comments and made the necessary revisions to improve the quality of the manuscript. A detailed point-by-point response to your comments has been provided in the attached Word document. For your convenience, we briefly summarize the key changes and responses here:

  1. Comment 1: Regarding the simplicity of the model architecture.
    Response: We have clarified the rationale behind the model design in Section 3.2 and highlighted the teacher model’s role in achieving high classification performance. Detailed responses are in the attached document (see Response 1).

  2. Comment 2: About the four-class classification framework.
    Response: We have explained that the SR category already represents healthy ECG signals and added clarification in Section 3.1 (lines 256–269). Detailed responses are in the attached document (see Response 2).

  3. Comment 3: On the comparative analysis.
    Response: We have explicitly discussed the comparison results in Section 4.3 (lines 514–526) and included references in Table 8. Detailed responses are in the attached document (see Response 3).

For the full responses with detailed explanations, please refer to the attached document. Thank you again for your thoughtful comments and suggestions.

Reviewer 2 Report

Comments and Suggestions for Authors

-          The authors need to claim this statement "the performance of arrhythmia classification models trained directly on single-lead ECG often fails to match the performance 78 levels of multi-lead models"

-          The authors should clearly specify the contribution of their work in the introduction.

-          The authors must clearly indicate the originality of this work.

-          It would be beneficial to discuss the limitations they found in lightweight approaches related arrhythmia classification and that they consider improving in their paper in terms of response time, precision, false positive and accuracy.

-          Incorporate a comparative table comparing existing works with the proposed approach.

-          The authors need to distinguish between hard labels and soft labels.

-          Figure 1 does not effectively demonstrate the impact of teacher model. The authors applied teacher model to extract more features (288) by considering LSTM and four ResNet blocks for deeper feature extraction, but the figure does not display how this increased the performance of student model. A clear representation of the before-and-after effect of teacher model on the classification of arrhythmia using student model is necessary. In addition, It would be helpful to provide details about 288 features.

-          The authors claim that using LSTM with four ResNet blocks enhance performance. In parallel the authors use two convolutional layers, one fully connected layer, and a Softmax output layer. However, results supporting this claim are not provided. An ablation study showing the performance of student model compared to the combined teacher-student models would help the reader understand the benefit of the hybrid approach  

-          It would be helpful to provide pseudocode and more details of teacher and student model.

-          The authors should evaluate and compare the computational complexity and execution time of the proposed approach and related approaches.

-          It would be valuable to provide a more detailed interpretation and analysis of the results.

-          The limitations of the study need more openly discussed.

Author Response

Dear Reviewer,

Thank you very much for your thorough review and valuable comments on our manuscript. We have carefully addressed all your suggestions and revised the manuscript accordingly. Below, we provide a brief summary of the main revisions made based on your comments:

  1. Comment 1: Regarding the performance of single-lead vs. multi-lead models.
    Response: We have added supporting evidence in the Introduction section (lines 79–83) to substantiate this statement.

  2. Comment 2: Specifying the contributions of this study.
    Response: We have explicitly highlighted the main contributions in the Introduction section (lines 91–103).

  3. Comment 4: Discussing the limitations of lightweight approaches.
    Response: We have added a detailed discussion of the limitations of pruning, quantization, and knowledge distillation in Section 2.3 (lines 208–221).

  4. Comment 8: Providing ablation study results.
    Response: We conducted an ablation study to evaluate the importance of ResNet blocks and LSTM layers in the teacher model. Results and analysis have been added to Section 3.2.1 (lines 345–373).

For a detailed point-by-point response, including all other comments and corresponding revisions, please refer to the attached document.

We sincerely appreciate your constructive feedback and suggestions, which have greatly contributed to improving the quality of our manuscript.

Round 2

Reviewer 1 Report

Comments and Suggestions for Authors

Nill

Reviewer 2 Report

Comments and Suggestions for Authors

The authors have answered all the questions put to them fully and comprehensively. The study has improved in all aspects of form and content and I recommend its acceptance.